# Early detection of complications in pancreas transplants by microdialysis catheters, an observational feasibility study

Gisle Kjøsen[1,2☯]*, Kristina Rydenfelt[1,2☯], Rune Horneland[3☯], Einar Martin Aandahl[3,4☯], Pål-Dag Line[2,3☯], Eric Dorenberg[5], Audun Elnæs Berstad[5], Knut Brabrand[5], Gaute Hagen[5], Sören Erik Pischke[1,6], Gisli Björn Bergmann[1¤], Espen Nordheim[2,3], Trond Geir Jenssen[2,3], Tor Inge Tønnessen[1,2‡], Håkon Haugaa[1,7‡]

1 Department of Anesthesiology, Oslo University Hospital, Oslo, Norway, 2 Institute of Clinical Medicine, University of Oslo, Oslo, Norway, 3 Department of Transplantation Medicine, Oslo University Hospital, Oslo, Norway, 4 Institute for Cancer Research, Oslo University Hospital, Oslo, Norway, 5 Department of Radiology and Nuclear Medicine, Oslo University Hospital, Oslo, Norway, 6 Department of Immunology, Oslo University Hospital, Oslo, Norway, 7 Lovisenberg Diaconal University College, Oslo, Norway

☯ These authors contributed equally to this work.
¤ Current address: Division of Anaesthesia and Intensive Care, Dept. of Operations, National University Hospital, Reykjavik, Iceland
‡ These authors also contributed equally to this work.
* gisle.kjosen@ous-hf.no

**Data Availability Statement:** The datasets generated and analysed during the current study are not publicly available due to Norwegian national

## Abstract

### Background

Despite advances in immunosuppression and surgical technique, pancreas transplantation is encumbered with a high rate of complication and graft losses. Particularly, venous graft thrombi occur relatively frequently and are rarely detected before the transplant is irreversibly damaged.

### Methods

To detect complications early, when the grafts are potentially salvageable, we placed microdialysis catheters anteriorly and posteriorly to the graft in a cohort of 34 consecutive patients. Glucose, lactate, pyruvate, and glycerol were measured at the bedside every 1–2 hours.

### Results

Nine patients with graft venous thrombosis had significant lactate and lactate–to-pyruvate-ratio increases without concomitant rise in blood glucose or clinical symptoms. The median lactate in these patients was significantly higher in both catheters compared to non-events (n = 15). Out of the nine thrombi, four grafts underwent successful angiographic extraction, one did not require intervention and four grafts were irreversibly damaged and explanted. Four patients with enteric anastomosis leakages had significantly higher glycerol measurements compared to non-events. As with the venous thrombi, lactate and lactate-to-pyruvate ratio were also increased in six patients with graft surrounding hematomas.

legislation prohibiting the publishing of information that could compromise research participant privacy. The collected data are of such nature that it falls under the provisions of the Norwegian Health Research Act. The data are available from Oslo University hospital's data protection official for research upon reasonable request, contact via personvern@ous-hf.no, for researchers who meet the criteria for access to confidential data.

**Funding:** This work was supported by an unrestricted research grant from South-Eastern Norway Regional Health Authority (2016028).

**Competing interests:** The authors have declared that no competing interests exist.

**Abbreviations:** ASA, Acetylsalicylic acid; AUC, Area under curve; BID, "bis in die", twice daily; CRP, C-reactive protein; CT, Computed tomography; DBD, Donation by brain death; DUS, Doppler ultrasound; IQR, Interquartile range; IU, International units; IVC, Inferior vena cava; LMWH, Low molecular weight heparin; NPV, Negative predictive value; PMP, Per million population; POD, Postoperative day; PPV, Positive predictive value; PTA, Pancreas transplant alone; ROC, Receiver operating characteristics; SC, Subcutaneously; SPK, Simultaneous pancreas-kidney.

## Conclusions

Bedside monitoring with microdialysis catheters is a promising surveillance modality of pancreatic grafts, but differentiating between the various pathologies proves challenging.

## Introduction

Pancreas transplantation has a high rate of complications such as graft venous thrombi, and exocrine leakages with subsequent infections and fistulae [1,2]. Monitoring the pancreatic transplants postoperatively remains challenging with the current standard of care, which includes vital signs, blood glucose and C-peptide, serum amylase, and the measurement of amylase in drained fluid. Doppler ultrasound (DUS), computed tomography [3] and magnetic resonance imaging (MRI) have all been utilized in the monitoring process, but the ideal radiological modality has yet to be defined [4–6]. In particular, venous transplant thrombi leading to ischaemia often remain undetected until the patient develops hyperglycaemia. At this stage, the grafts are typically beyond salvation, although some grafts may be saved upon immediate reperfusion. Tools which allow earlier detection of circulatory complications in the transplant may therefore improve graft survival.

Microdialysis is a well-known monitoring technique of tissues and organs [7]. The catheters consist of one inlet and one outlet canal connected by a semipermeable membrane at the tip (Fig 2). This allows transport of metabolites like glucose, lactate, pyruvate, and glycerol from the tissue into the catheter for subsequent bedside analysis, which enables close to real time monitoring of tissue metabolism [8–11]. At our institution we routinely monitor paediatric liver transplants with microdialysis catheters [12].

At Oslo University hospital the incidence of venous thrombi in pancreas grafts was reported to be 23% in 2015, and an incidence of graft loss due to venous thrombi of 9%. The aim of this study was to investigate the microdialysis method´s potential to detect venous thrombi and other common complications after pancreas transplantation. Our hypothesis was that by microdialysis monitoring, complications such as graft venous thrombosis can be detected earlier than by the current standard of care. Ultimately, this could provide rapid intervention which in turn may increase graft survival.

## Methods

### Study design and patient population

The study was approved by the South-Eastern Norwegian ethical committee and registered as a sub study of The Norwegian Pancreas Transplantation (PTx) Study at www.clinicaltrials.gov (NCT01957696). Written informed consent was obtained from all participants.

All solid organ transplantation procedures in Norway are performed at Oslo University Hospital. This investigation was designed as an open-label observational feasibility study. All organ donations during this time period were by means of "donation by brain death" (DBD). Between May 2015 and July 2016, thirty-four consecutive pancreas transplant recipients, who either received a pancreas transplant alone [13] or a simultaneous pancreas kidney transplantation (SPK) were included. All 34 patients had microdialysis catheters implanted. See Consort flow diagram (Fig 1). The only eligibility criteria were a planned PTA/SPK procedure, and the ability to give informed consent. Inclusion was performed at admission for the transplantation. No incentives to adhere to protocol were offered.

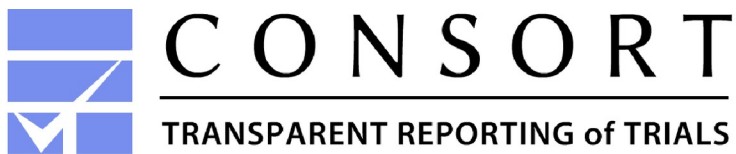

## CONSORT 2010 Flow Diagram

**Enrollment**

Assessed for eligibility (n= 34)

Excluded  (n=0)
- Not meeting inclusion criteria (n=0)
- Declined to participate (n= 0)
- Other reasons (n=0)

Randomized (n= 34)

**Allocation**

Allocated to intervention (n= 34)
- Received allocated intervention (n=34)
- Did not receive allocated intervention (n=0)

Allocated to intervention (n=0)
- Received allocated intervention (n=0)
- Did not receive allocated (n= 0)

**Follow-Up**

Lost to follow-up (n=0)

Discontinued intervention (n= 0)

Lost to follow-up (n= 0)

Discontinued intervention (n= 0)

**Analysis**

Analysed  (n=34)
- Excluded from analysis (n= 0)

Analysed  (n=0)
- Excluded from analysis (n= 0)

**Fig 1. Consort flow diagram.** The Consort flow diagram for the study. Template sourced from https://www.equator-network.org/wp-content/uploads/2013/09/CONSORT-2010-Flow-Diagram-MS-Word.doc.

## Surgical procedure

The pancreas grafts were transplanted through a midline incision and placed "standing" in the right retrocolic position. The pancreatic arteries were anastomosed to the right common iliac artery, and the portal vein was anastomosed to the lowermost inferior caval vein. The enteric anastomosis was by duodenoduodenostomy [14].

Before closure of the abdominal incision, arterial flow was measured in the superior mesenteric artery and the coeliac trunk, using a doppler flow probe (Medistim VeriQ, Medistim ASA, Oslo, Norway).

## Microdialysis

At the end of the surgical procedure one microdialysis catheter (CMA 65, M Dialysis AB, Stockholm, Sweden) was sutured to the anterior pancreatic surface, and one on the posterior surface (Fig 2). The catheters were perfused with 6% hydroxyethyl starch, using a microdialysis pump (CMA 107, M Dialysis AB), at a rate of 1 μL/min [15]. The samples were analysed for glucose, lactate, pyruvate and glycerol every hour for the first 24 hours, and every 2–3 hours

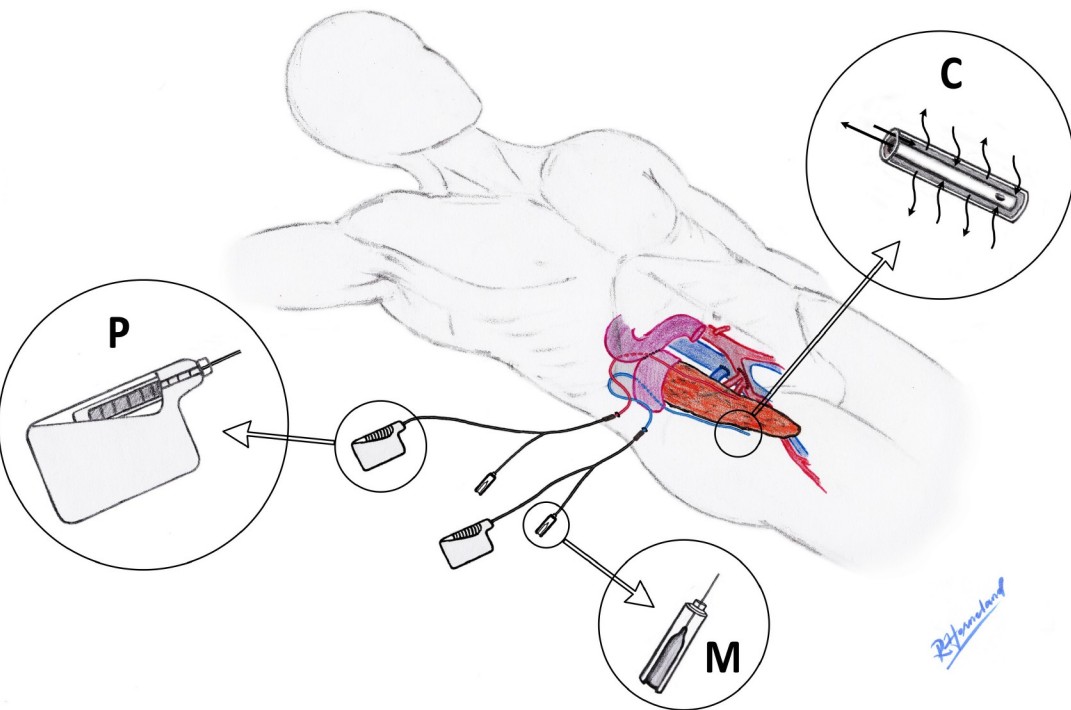

**Fig 2. The microdialysis setup and placement.** Microdialysis catheters positioned anterior and posterior of a pancreas transplant. The common aortic patch with both the coeliac trunk and the superior mesenteric artery was anastomosed end-to-side to the right common iliac artery. The portal vein with an iliac vein interposition graft from the same donor was anastomosed "end-to-side" to the lowermost inferior caval vein. The enteric anastomosis was performed as a hand-sewn side-to-side duodenoduodenostomy. P: Battery driven syringe pump perfusing 6% hydroxyethyl starch at a rate of 1 μL/min. C: Catheter tip with a 30 mm long semipermeable membrane with a molecular weight cutoff of 100 kDa. M: Microvial holder with changeable microvial/bottle.

thereafter (Iscus Flex, M Dialysis AB). Lactate-to-pyruvate [16] ratio was calculated. The catheters have a priming volume of 7.2μL. They have a tip-to-vial sampling delay of 8 minutes, when using a flow rate of 1 μL/min. The catheters were kept as long as they were able to adequately sample, or until the patient was discharged from the surgical department.

According to our previous experience with other solid organ transplants we surmised that a 50% increase in lactate compared with the previous measurement was indicative of a pathological process [9,12,17]. If the increase was confirmed by a repeat measurement 30 minutes later, a DUS or CT scan was performed within one hour. The choice of modality was left to the discretion of the transplant surgeon on-call.

## Imaging

According to our clinical protocol, all patients underwent a routine DUS within four hours after reperfusion, and a CT scan was only performed if the DUS was inconclusive. From the ninth patient in the study and onwards, DUS and contrast enhanced CT scan (Iomeron, Bracco, Milan, Italy) were performed between postoperative day (POD) five and seven according to protocol. Additional DUS and CT were performed on clinical indication.

## Thrombosis and infection prophylaxis

Before clamping the iliac artery, 30 IU/kg unfractionated heparin was administered intravenously. Six hours postoperatively 2500 IU low molecular weight heparin (LMWH) was administered subcutaneously. Thereafter, LMWH 2500–7500 IU sc daily was administered, depending on patient weight, uremic status, and type of transplant. Acetylsalicylic acid (ASA) 75 mg was administered daily from POD seven. For infection prophylaxis, meropenem, vancomycin and micafungin were administered intravenously intraoperatively.

## Immunosuppression

All patients received similar immunosuppressive therapy which included an induction protocol consisting of 5–7 mg/kg anti-thymocyte globulin (ATG), in addition to standard maintenance immunosuppression with tacrolimus, Mycophenolate Mofetil (MMF) and corticosteroids.

## Statistical analysis

For sample size calculation, we used Power Analysis (A& X Analytics, LLC) adjusted for later test with an unpaired two sample two tail t-test. With an incidence of venous thrombosis of 10%, a normal lactate value of 1.5 mM vs. a pathological value of 2.5 mM (95% confidence interval +/- 0.7 mM), power of 0.80, and an alpha level of 0.05, we calculated that a minimum of 42 patients was needed. A pathological lactate level of 2.5mM was chosen based on our previously published results [8,9].

Inter-group comparisons between donors, recipient- and graft characteristics were performed with the Mann-Whitney-*U*-test. To conservatively report group differences in this observational feasibility study, we used values obtained by the microdialysis catheters during the entire observational period for group comparisons. For repeated measurements and comparisons of the two catheters the Wilcoxon-Signed-Rank test was used. Using data from the group comparison analyses, the ability of the measured variables to discriminate grafts with venous thrombi from those with uneventful clinical courses was explored with receiver operating characteristics (ROC). The area-under-curve (AUC) was calculated. The optimal cutoff-value for each variable was defined as the value closest to the top left corner. We report the

values for sensitivity and specificity yielded from the ROC-curve analyses. Optimal cutoff-values were used to calculate positive and negative predictive values (PPV and NPV) in contingency tables. As opposed to the ROC curves where the thrombi cohort was compared to the non-event cohort only, all patients were included in the contingency table analyses. All P-values presented are 2-tailed and values <0.05 were regarded as statistically significant. Statistical analysis was conducted using IBM SPSS Statistics for Windows, version 26 (IBM, Armonk, NY, USA).

## Results

Recipient and donor characteristics of the included patients are presented in Table 1. There were no patients lost to inclusion, no withdrawals from the study, and none lost to follow up. All 34 patients were included in the analysis.

The study was prematurely terminated after 34 patients due to a revision of the transplantation protocol instigated by an increased incidence of venous thrombi in the cohort.

**Table 1. Donor and patient characteristics in 34 consecutive pancreas transplantations, all transplantations and by event type.**

| | All events | References (n = 15) | Venous thrombi (n = 9)[1] | | Enteric Leakages (n = 4) | | Haematomas (n = 6) | |
|---|---|---|---|---|---|---|---|---|
| **Recipient** | | | | $p^2$ | | $p^2$ | | $p^2$ |
| Patients (n) | 34 | 15 | 9 | | 4 | | 6 | |
| SPK, n (%) | 18 (52.9) | 11 (73.3) | 3 (33.3) | | 1 (25) | | 3 (50) | |
| PTA, n (%) | 16 (47.1) | 4 (26.7) | 6 (66.7) | | 3 (75) | | 3 (50) | |
| Age (IQR) | 42 ± 8.7 | 39.6 ± 7 | 43.9 ± 5.7 | .174 | 46.3 ± 6.5 | .124 | 40.7 ± 5.8 | .791 |
| Height (IQR) | 173.5 (167.6–180) | 169 (166–175) | 173 (169–182) | .215 | 175 (169.5–179.8) | .53 | 179.5 (176.8–180) | .132 |
| Weight (IQR) | 71.5 (62.8–86.3) | 65.5 (61–73.5) | 87 (68–92) | .174 | 75.9 (65.4–83.3) | .596 | 74 (67–76.7) | .381 |
| Recipient BMI (IQR) | 23.6 (21–27.8) | 22.9 (20.7–26.1) | 28.4 (20.5–29.4) | .318 | 23.1 (21.5–25.4) | .81 | 23 (21.4–23.7) | .85 |
| Recipient diabetes duration, years | 27 ± 10 | 28.1 ± 7.9 | 25.6 ± 9.3 | .558 | 23.5 ± 12.1 | .665 | 30.3 ± 10.5 | .622 |
| Ischemia time, hours: minutes (IQR) | 8:14 (6:24–11:05) | 09:32 (07:51–12:32) | 06:24 (05:56–08:03) | .03* | 10:05 (08:33–11:20) | .99 | 07:07 (06:14–08:43) | .045* |
| Total pancreatic arterial blood flow after reperfusion, ml/min (IQR)[3] | 200 (163–258) | 245 (205–341) | 168 (115–192) | .01* | 175 (161.3–208.8) | .08 | 145 (140–187.5) | .003* |
| Duration of microdialysis measurements (days) (IQR) | 7 (5–9) | 8 (5–9) | 5 (2–8) | .215 | 6 (6–6.8) | .665 | 10.5 (7–11.8) | .154 |
| Sex, male/female (%) | 21/13 (62/38) | 9/6 (60/40) | 5/4 (56/44) | | 3/1 (75/25) | | 4/2 (67/33) | |
| **Donor** | | | | | | | | |
| Age | 29.4 ± 12.8 | 26.5 ± 12.4 | 32.1 ± 11.9 | .29 | 28.8 ± 15.2 | .81 | 33.2 ± 11.2 | .34 |
| Height | 172.5 (165.5–180) | 170 (159.5–180) | 175 (167–180) | .446 | 175 (164.5–185.3) | .411 | 177.5 (175–184.5) | .55 |
| Weight | 72 (60.5–80) | 70 (60–75.5) | 72 (65–80) | .519 | 70 (60–80) | .885 | 81.5 (75.8–88.8) | .55 |
| BMI | 24.3 (22.3–26.2) | 25 (21.9–26.4) | 24.7 (22.5–25.7) | .77 | 22.9 (22.4–23.2) | .307 | 26 (23–27.8) | .381 |
| Sex, male/female (%) | 16/18 (47/53) | 5/10 (33/67) | 4/5 (44/56) | | 3/1 (75/25) | | 4/2 (67/33) | |

BMI: Body mass index; IQR: Interquartile range; PTA: Pancreas transplant alone; SD: Standard deviation; SPK: Simultaneous pancreas kidney.

[1] Eight primary venous thrombosis and one secondary to fungal infection.

[2] Comparison of the pathological cohorts vs. the reference cohort (Mann Whitney *U* test).

[3] Measured intraoperatively.

* denotes statistical significance.

**Table 2. Laboratory values from the entire observational period (median (quartiles)) in 34 consecutive pancreas transplants.**

|  | References (n = 15) | Venous thrombi (n = 9)[1] |  | Enteric leakages (n = 4) |  | Hematomas (n = 6) |  |
|---|---|---|---|---|---|---|---|
|  | Median | Median | *P*[2] | Median | *P*[2] | Median | *P*[2] |
| **Glucose (mM)** | 5.8 (5.1–6.9) | 7.3 (6.1–8.4) | *.008** | 6,1 (5.2–7.3) | .60 | 6.1 (5.3–7.0) | .52 |
| **Lactate (mM)** | 1.4 (0.9–1.9) | 1.1 (0.9–1.8) | .51 | 1.0 (0.7–1.3) | *.027** | 1.2 (0.8–1.4) | .21 |
| **CRP (mg/L)** | 19 (10–38) | 56 (29–68) | *.025** | 62 (41–86) | *.004** | 49 (24–101) | *.023** |
| **Serum Amylase (U/L)** | 46 (29–66) | 52 (39–55) | .73 | 42 (34–90) | .81 | 71 (47–102) | .21 |
| **Creatinine (μM)** | 109 (78–191) | 87 (69–143) | .52 | 67 (48–106) | .19 | 117 (91–141) | .85 |
| **Urea (mM)** | 6.4 (4.6–10.3) | 5.3(4.4–9.6) | .86 | 5.2 (4.1–6.3) | .22 | 6.4 (4.9–10.1) | .47 |
| **Drain amylase (U/L)** | 89 (30–244) | 179 (152–659) | *.142* | 394 (69–2693) | .26 | 111 (54–228) | .91 |

[1]Eight primary venous thrombosis and one secondary to fungal infection.

[2]Comparison of the pathological cohorts vs. the reference cohort (Mann Whitney *U* test).

* denotes statistical significance.

Of the 34 patients included, nine developed venous thrombi, four developed leakages in relation to the enteric anastomosis, and six patients experienced bleeding/hematomas surrounding the pancreas graft. A total of four grafts were lost, all due to thrombi. Fifteen patients had no events and served as a reference cohort. These patients had stable blood glucose levels (median 5.8 mM (interquartile range (IQR) 5.1–6.9) without any need of insulin, a median serum-amylase of 46 U/L (IQR 29–66) and low amylase-concentrations in drained fluid (median 89 U/L (IQR 30–224). All laboratory values are presented in Table 2.

A subgroup analysis with regards to these four cohorts revealed a statistically significant lower total pancreatic blood flow, and a shorter cold ischaemia time in the thrombus group versus the reference cohort. The same was found for the haematoma group versus the reference cohort. The details are presented in Table 1.

There were no adverse events or complications related to the insertion of the microdialysis catheters. The catheters had a median functional duration of seven days (IQR 5–9). The reasons for the cessation of microdialysis monitoring were loss of functional sampling (n = 17), transfer of the patient from the surgical- to the medical ward (n = 14), and removal of catheters in conjunction with organ explantation (n = 3). For the 17 catheters that ceased functioning, the median catheter life was 5.5 days for the anterior catheter (range 1–13), and 5 days for the posterior catheter (range 1–13). All catheters were removed without complications. There was no follow-up after the removal of the catheters.

## Venous thrombi

In the absence of any clinical symptoms or standard laboratory findings, the microdialysis measurements showed a marked increase in lactate for all the nine venous thrombi which lead to detection of a thrombus in seven cases. The presence of a thrombus was confirmed by contrast enhanced CT or DUS. In one patient with an increase in microdialysis lactate during the first 24 postoperative hours and unremarkable postoperative DUS, the presence of a thrombus was first diagnosed by the protocol CT-examination at POD six. In one patient, the microdialysis analyser failed bedside. This patient developed a thrombus that led to irreversible ischemia, necessitating a graftectomy within the first 24 hours. These two thrombi were by retrospective analysis readily detectable by microdialysis measurements by an increased lactate and lactate to pyruvate ratio. Of the four thrombi resulting in graft loss, two were detected with delay due to a malfunction of the analyser, the other two were erroneously interpreted as

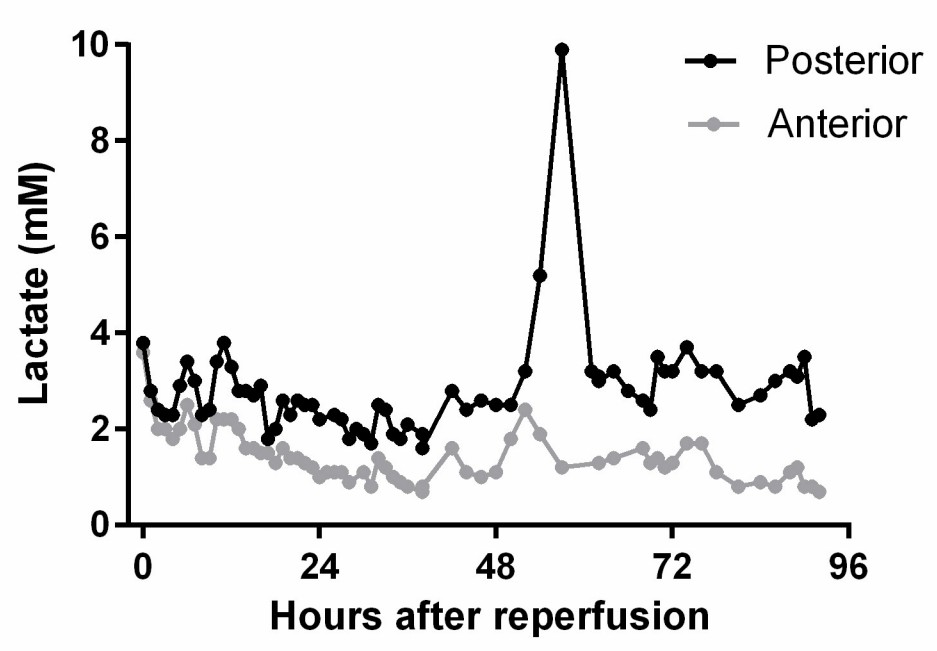

**Fig 3. Microdialysis lactate elevation.** Lactate concentration in a pancreas graft with venous thrombus, measured in microdialysate sampled posterior and ventral to the transplanted pancreas. Notice the peak in the posterior lactate level, followed by a rapid fall after circulation was re-established by thrombus extraction.

having normal values in the start-up process of ascertaining the normal levels in pancreatic transplantation.

The increase in lactate was detected between three hours post reperfusion and up to POD-five with a median of 28 hours (IQR 5.5–42), (example in Fig 3). The median time from the first elevation in microdialysis lactate to the first diagnostic CT (n = 6, example in Fig 4) or DUS (n = 1) was 4.1 hours (IQR 3.6–7.9 hours). The nine patients with venous thrombi had a median peak microdialysis lactate-value of 6.4 mM (IQR 5.5–6.4), and during the entire observation period they had higher lactate values compared to non-event patients (Table 3, example in Fig 3). The lactate-to-pyruvate-ratio was significantly higher in the patients with venous thrombi.

The AUC value for lactate discriminating thrombotic grafts from uneventful grafts was 0.88 (95% CI 0.74–1.00, $p$ = 0.003) for the anterior catheter and 0.82 (CI 0.62–1.00, $p$ = 0.011) for the posterior catheter. Based on the ROC-curve, the anterior catheter cutoff value was 1.7 mM, giving a sensitivity of 88% and a specificity of 87% to detect a thrombus. For the posterior catheter, a cutoff lactate-concentration of 2.4 mM yielded a sensitivity of 78% and a specificity of 87%. The optimal cutoff values obtained from the ROC-curves revealed PPV-values for the entire cohort with a median of 31% (range 24–40%) with no improvement by repeated measurements, simultaneous elevation in both catheters, or combining lactate and lactate-to-pyruvate ratio. The NPV values followed the same pattern with a median value of 82% (range 56–93%) (Table 4).

In the thrombus cohort the blood glucose levels were near identical to the non-event cohort, with a median of 5.8 (IQR 5.2–7.2, $p$ = 0.770) prior to detection of a thrombus. The blood glucose was only slightly elevated during ischemia, as compared to the pre-thrombus level, with a median of 6.4 mM (IQR 5.8–7.9, $p$ = 0.052). No inter-group differences in the

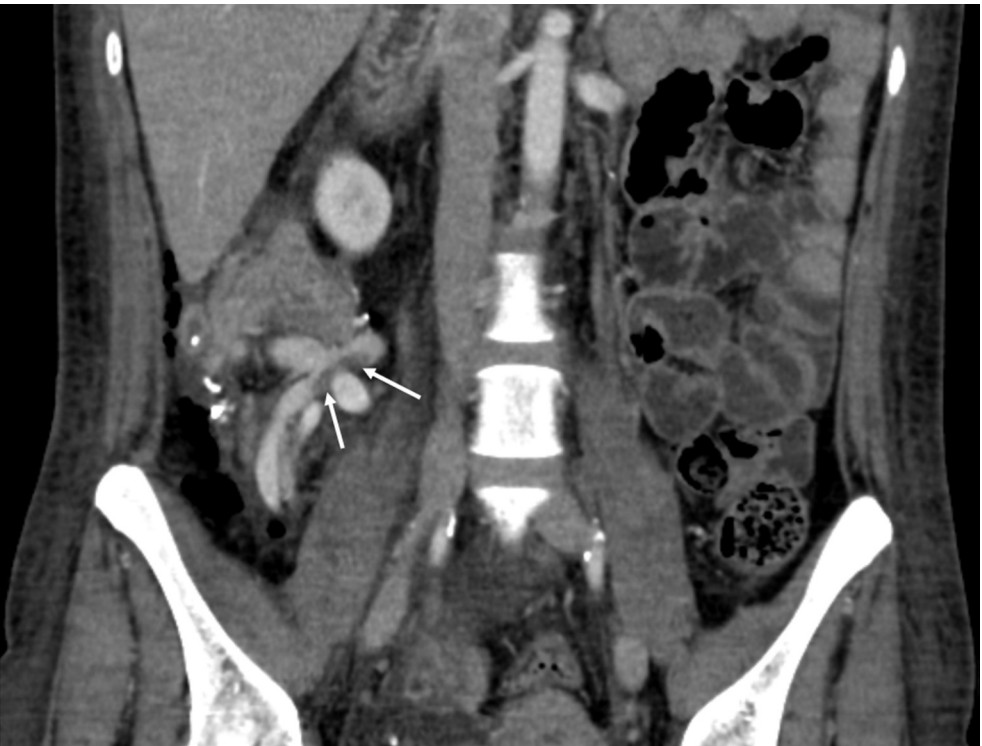

**Fig 4. Thrombus on CT scan.** Computed tomography scans with intravenous contrast in venous phase of a pancreas transplanted patient showing a thrombus in the portal segment of the graft vein.

standard laboratory measured biochemical markers were seen, other than a modest CRP increase in all pathological cohorts, and a minimally elevated serum lactate in the leakage cohort (Table 3).

DUS examination revealed the presence of a thrombus in only two out of nine patients, the contrast enhanced CT in eight. In the 9th patient, a CT scan was not performed as the DUS findings were conclusive. The CT scan confirmed the presence of all hematomas and leakages.

Four grafts that developed venous thrombi, all detected by microdialysis surveillance, were rescued by means of angiographic extraction of the thrombus. All remained insulin free for more than a year post thrombus extraction.

**Table 3. Microdialysis values from the entire observational period (median (quartiles)) from catheters placed anteriorly and posteriorly of 34 consecutive pancreas transplants.**

| | References (n = 15) | Venous thrombi (n = 9)[1] | | Enteric leakages (n = 4) | | Hematomas (n = 6) | |
|---|---|---|---|---|---|---|---|
| | Median (ant & post) | Median (ant & post) | $P^2$ | Median (ant & post) | $P^2$ | Median (ant & post) | $P^2$ |
| **Glucose (mM)** | 4.8 (4.7–5.5) | 4.7 (4.0–5.2) | .42 | 5.2 (3.7–6.1) | .060 | 4.1 (3.6–4.3) | .005* |
| **Lactate (mM)** | 1.4 (1.1–1.8) | 2.9 (2.6–3.3) | .005* | 3.0 (2.1–4.2) | .02* | 3.2 (2.7–4.0) | < .001 |
| **Pyruvate (μM)** | 112 (95–121) | 136 (130–176) | .003* | 130 (117–136) | .15 | 142 (126–152) | .011 |
| **Lactate/pyruvate** | 13 (11–14) | 18 (14–19) | .004* | 23 (17–30) | .036* | 24 (21–26) | .002 |
| **Glycerol (μM)** | 58 (46–145) | 114 (62–265) | .19 | 597 (276–884) | .004* | 63 (58–214) | .34 |

[1]Eight primary venous thrombosis and one secondary to fungal infection.

[2]Comparison of the median values in the pathological cohorts vs. the reference cohort (Mann Whitney *U* test).

* denotes statistical significance.

### Enteric leakages

Four patients developed CT confirmed leakages in relation to the enteric anastomosis. Compared to non-event patients, these patients had a significantly higher level of glycerol in the microdialysis samples (Table 2). The elevation in glycerol preceded any clinical signs and symptoms by up to 14 days. The microdialysis lactate levels were also significantly higher than in the non-event patients. The median drain amylase for the leakage group was 394 U/L (IQR 69–2693), as compared to 89 U/L (IQR 30–244) in the reference cohort ($p = 0.26$). There were no other differences in the laboratory values.

### Hematomas

Six patients developed bleeding/hematomas in relation to their pancreas graft verified by CT. The patients had significant increases in microdialysis lactate, pyruvate, and lactate-to-pyruvate ratio compared to non-events, whereas the microdialysis glucose levels were lower (Table 2). The hematomas were detected in the early postoperative phase (median time 20:30 hrs. (IQR 17:45–32:15 hrs.). All six patients had a marked decrease in blood haemoglobin, with a median decline from a pre-bleed level of 12.3 (IQR11.8–14.6) g/dL down to a median low of 7.8 (IQR 7.2–8.7) g/dL ($p = 0.028$).

## Discussion

To our knowledge this is the first report of the microdialysis method being utilized in the setting of pancreas transplantations. Our main finding was that the microdialysis catheters detected venous graft thrombosis at an early stage with high sensitivity and specificity compared to an uneventful cohort. Complications were detected before clinical symptoms, or any elevation in biochemical markers, supporting our hypothesis that common complications can be detected at an earlier time point than current monitoring tools. We have shown by examples, that the detection of ischemia rapidly instigated further diagnostic measures and definitive treatment of venous thrombosis. However, an elevated lactate solely was not accurate as a diagnostic tool due to the multiple aetiologies of lactate formation. Our results also show that microdialysis detects exocrine and enteric pancreas graft leakages, that may subsequently lead to infections, at an early stage.

Microdialysis can be applied to monitor liver transplants [9,12,18]. The most important difference in sampling from liver- versus pancreas grafts is that liver grafts allow for safe and easy intrahepatic sampling. Due to the risk of leakage we chose to place the catheters as close to the pancreatic surface as possible. Our hypothesis was that small molecules like lactate, pyruvate, glycerol, and glucose would permeate through the pancreatic capsule, and thus render them available for sampling by the catheters. The data suggest that this may be correct, but their relation to intraparenchymal levels is uncertain.

The time from formation of an occluding thrombus to the reestablishment of blood flow is crucial in graft survival. The rationale for closer monitoring of pancreas grafts is to detect ischemia *before* the thrombus has reached the size where it completely occludes the venous drainage and endangers the survival of the graft. This can potentially result in interventions performed on subclinical thrombi that could be successfully managed by anticoagulation alone. Alternatively, the grafts were salvageable *because* the thrombi did not fully occlude the vein. Given the dismal outcome of total graft venous occlusion and the relatively low complication rate of interventional angiography we find this approach justifiable [19].

The low PPV for lactate in venous thrombi reflects the multiple mechanisms of lactate generation. Enteric leakages trigger an inflammatory response in the pancreas and surrounding tissue which then leads to an elevated lactate [20,21]. Erythrocytes produce lactate as a part of

their metabolism; therefore, hematomas will also confound the interpretation of lactate values [22]. The lower level of microdialysis glucose in the hematoma group was also likely due to the ongoing metabolism of the extra-circulatory erythrocytes and leukocytes within.

Leakages originating from the duodenoduodenostomy and/or from the pancreas graft itself were detected at very early time points in our study. This data is in accordance with a previous report on patients with pancreatic cancer undergoing pancreatoduodenectomy (Whipple's procedure) [20]. Leakages from the pancreatic anastomosis can be detected by increased glycerol in microdialysate collected at the pancreaticojejunostomy well in advance of clinical signs and symptoms.

The three pathological event groups and the reference cohort were similar in both donor and recipient baseline characteristics. There were however statistical differences in the thrombus cohort versus the control group with a notable shorter cold ischaemia time. It is well established that a prolonged ischaemia time places the organ at a greater risk of adverse events [16,23], but in this study the opposite was found. This study by itself, however, is not sufficient to challenge this previously established relationship. The same difference in cold ischaemia time was found in the haematoma subgroup, but again there is little to support any causality between a shorter ischaemia time and an increased risk of adverse events. Both groups have few patients, and care should be exercised in drawing conclusions solely based on these findings.

There were, however, also statistical differences between the thrombi group and the reference cohort as well as the haematoma group and the reference cohort with regards to the total pancreatic blood flow measured intraoperatively, which was significantly lower in both groups. As one of the three factors comprising Virchow's triad, the association between stagnant blood flow and thrombus formation is well established. The measurements were however arterial, and the thrombi were venous. At our institution we do not routinely measure venous blood flow intraoperatively in pancreas transplantations. Likewise, do we not record the arterial flow pattern. An absent or reversed arterial flow pattern could represent an increased venous resistance, for instance due to a thrombus [24]. But without a recorded arterial flow pattern this explanation is difficult to verify. As we only assessed arterial blood flow, we can only speculate that low arterial blood flow leads to more stagnant venous flow and thus a higher risk for thrombus formation.

The association between diminished arterial blood flow measured intraoperatively, and an increased incidence of bleeding is somewhat difficult to explain. A possible pathophysiological explanation could be that a venous occlusion could lead to an engorged organ and diminished arterial flow, and secondary to this an increased risk of bleeding. However, in this study, all CT scans for the six patients with haematomas showed open arteries and veins.

In the design of the study, we chose to use two microdialysis catheters to increase the duration of monitoring in case of catheter dysfunction, and to assess any potential differences between the two catheters. We experienced patients whose pathology was contained to one catheter alone, with normal values in the contralateral catheter. The most likely explanation is that the microdialysis catheters sample *locoregionally*, rather than the organ as a whole. This partially explains the low PPV and NPV values derived from both catheters. The catheters were sutured in place, but the position of the catheter tips in relation to the grafts may have varied over time and that some of the catheters may sampled too far away from the graft.

There is a possibility that we, by close monitoring, have detected venous thrombi that otherwise may have had a subclinical course if left untreated. The graft loss rate of 4/34 (12%) is comparable to our previous reports [14,25], but higher than the 4% venous thrombosis rate reported by Walter et al in 2014 [26] as well as previously published figures from our institution [14]. The high proportion of PTA to SPK (50%) in this study, may in part explain the

**Table 4. Contingency table analyses showing how repeated measurements and catheter placements detected graft venous thrombosis in 34 pancreas transplants of which 9 had venous thrombosis[1].**

| Lactate >1.7 mM in the anterior and 2.4 mM in the posterior catheter | Pancreas transplant venous thrombosis[1] | | | |
|---|---|---|---|---|
| | Sensitivity | Specificity | Positive predictive value | Negative predictive value |
| *Single time point measurement* | | | | |
| Anterior catheter | 88.9 | 12.0 | 26.7 | 75.0 |
| Posterior catheter | 88.9 | 12.0 | 26.7 | 75.0 |
| Both catheters simultaneously | 66.7 | 24.0 | 24.0 | 66.7 |
| *Two consecutive measurements* | | | | |
| Anterior catheter | 88.9 | 16.0 | 27.6 | 80.0 |
| Posterior catheter | 88.9 | 16.0 | 27.6 | 80.0 |
| Both catheters simultaneously | 66.7 | 36.0 | 27.3 | 75.0 |
| *Three consecutive measurements* | | | | |
| Anterior catheter | 88.9 | 28.0 | 30.8 | 88.9 |
| Posterior catheter | 77.8 | 44.0 | 33.3 | 77.8 |
| Both catheters simultaneously | 55.6 | 52.0 | 29.4 | 55.6 |
| **Lactate >1.7 mM in the anterior and 2.4 mM in the posterior catheter, Lactate/pyruvate > 13.3 in the anterior and >19.1 in the posterior catheter** | | | | |
| *Single time point measurement* | | | | |
| Anterior catheter | 88.9 | 28.0 | 30.8 | 87.5 |
| Posterior catheter | 88.9 | 44.0 | 36.4 | 91.7 |
| Both catheters simultaneously | 66.7 | 60.0 | 37.5 | 83.3 |
| *Two consecutive measurements* | | | | |
| Anterior catheter | 88.9 | 32.0 | 32.0 | 88.9 |
| Posterior catheter | 88.9 | 52.0 | 40.0 | 92.9 |
| Both catheters simultaneously | 66.7 | 64.0 | 40.0 | 84.2 |
| *Three consecutive measurements* | | | | |
| Anterior catheter | 88.9 | 36.0 | 33.3 | 90.0 |
| Posterior catheter | 77.8 | 56.0 | 38.9 | 87.5 |
| Both catheters simultaneously | 55.6 | 68.0 | 38.5 | 81.0 |

[1] Pancreas transplant venous thrombosis verified by computed tomography examination.

increased complication rate compared to other publications, as PTA has previously been shown to have a higher complication rate and an inferior graft and patient survival [2,27–29].

In the design of the study, and based on our previous experiences, we opted to perform a repeat microdialysis measurement after 30 minutes to confirm or refute the presence of a pathological process, and thus minimise the risk of a spurious measurement. One could argue that the diagnosis of a potential calamitous process could be more rapidly diagnosed by an immediate DUS or CT scan after only one measurement. As shown in Table 4, the difference between one or two repeated measurements on positive predictive value, is negligible. However, the same negligible difference also speaks against relying on one measurement alone, especially when taking into consideration the increase in imaging this would lead to. Most of the radiological imaging undertaken would inevitably be without findings, and the usage of iodising contrasting agents would place the recipients at an increased risk of complications.

The study was terminated prematurely due to an unforeseen increase in the incidence of venous thrombi. This increase led to a revision of the transplant protocol, which amongst other adjustments involved increasing the routine anticoagulation administered to pancreas recipients. Continuing the study would potentially present a significant source of bias, we

therefore chose to cease further inclusion. There was no indication that the increase in thrombi was related to the microdialysis catheters.

The most critical *limitation* of this feasibility report is the limited number of patients. The study was terminated prematurely after 34 patients due to an unforeseen increased incidence of venous thrombi, necessitating a clinical protocol change. Continuing the study would have imbued a potential source of bias, since the changes made to the protocol included increased anticoagulation. The increased incidence was not suspected to be related to the microdialysis catheter monitoring due to their extraparenchymal placement.

Also, by utilizing microdialysis values from the entire observational period in all patients we may have underestimated differences in measurements between the pathological and reference cohorts. On the other hand, by doing so, we have assured not over-reporting any differences.

## Conclusion

The combination of early detection of graft ischemia with the microdialysis method, accurate diagnosis by CT, and extraction of venous thrombi by means of angiographic intervention seems feasible and may contribute to improving pancreas graft survival rates. However, at present, the metabolic parameters obtained by extra-pancreatic microdialysis catheters cannot differentiate between venous graft thrombosis, haemorrhage, and pancreatic leakages after pancreas transplantation. Future studies should be performed to address optimization of the catheter positioning, the possibility of intraparenchymal sampling, and if condition specific biomarkers can be obtained by the catheters.

## Supporting information

**S1 File. Clinical study protocol.** The protocol for the project, in which the microdialysis part is described in part 2.7.
(DOC)

**S2 File. TREND checklist.** Trend checklist for the study. Template sourced from https://www.cdc.gov/trendstatement/pdf/trendstatement_trend_checklist.pdf.
(PDF)

## Acknowledgments

### Declarations

The authors would like to thank the following for their support and assistance during the study:

Nurses in the operating theatre and the transplant surgical ward, in particular Kari G Dahl, MN.

Bioengineer Camilla Schjalm and supporting staff at the Institute of Immunology.

Medical technical staff, especially Anders Johnsen, for swift maintenance and repairs of equipment.

## Author Contributions

**Conceptualization:** Gisle Kjøsen, Kristina Rydenfelt, Sören Erik Pischke, Tor Inge Tønnessen, Håkon Haugaa.

**Data curation:** Gisle Kjøsen, Kristina Rydenfelt, Rune Horneland, Einar Martin Aandahl, Pål-Dag Line, Eric Dorenberg, Audun Elnæs Berstad, Knut Brabrand, Gaute Hagen, Espen Nordheim, Trond Geir Jenssen, Håkon Haugaa.

**Formal analysis:** Gisle Kjøsen, Kristina Rydenfelt, Håkon Haugaa.

**Funding acquisition:** Tor Inge Tønnessen, Håkon Haugaa.

**Investigation:** Gisle Kjøsen, Kristina Rydenfelt, Rune Horneland, Einar Martin Aandahl, Pål-Dag Line, Eric Dorenberg, Audun Elnæs Berstad, Knut Brabrand, Gaute Hagen, Sören Erik Pischke, Gisli Björn Bergmann, Espen Nordheim, Trond Geir Jenssen, Tor Inge Tønnessen, Håkon Haugaa.

**Methodology:** Gisle Kjøsen, Kristina Rydenfelt, Sören Erik Pischke, Gisli Björn Bergmann, Tor Inge Tønnessen, Håkon Haugaa.

**Project administration:** Gisle Kjøsen, Tor Inge Tønnessen, Håkon Haugaa.

**Resources:** Gisle Kjøsen, Kristina Rydenfelt, Rune Horneland, Sören Erik Pischke, Gisli Björn Bergmann, Tor Inge Tønnessen, Håkon Haugaa.

**Supervision:** Tor Inge Tønnessen, Håkon Haugaa.

**Validation:** Gisle Kjøsen, Kristina Rydenfelt, Rune Horneland, Einar Martin Aandahl, Pål-Dag Line, Sören Erik Pischke, Gisli Björn Bergmann, Tor Inge Tønnessen, Håkon Haugaa.

**Visualization:** Gisle Kjøsen, Kristina Rydenfelt, Rune Horneland, Einar Martin Aandahl, Pål-Dag Line, Sören Erik Pischke, Gisli Björn Bergmann, Tor Inge Tønnessen, Håkon Haugaa.

**Writing – original draft:** Gisle Kjøsen, Håkon Haugaa.

**Writing – review & editing:** Gisle Kjøsen, Kristina Rydenfelt, Rune Horneland, Einar Martin Aandahl, Pål-Dag Line, Eric Dorenberg, Audun Elnæs Berstad, Knut Brabrand, Gaute Hagen, Sören Erik Pischke, Gisli Björn Bergmann, Espen Nordheim, Trond Geir Jenssen, Tor Inge Tønnessen, Håkon Haugaa.

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
