## [Decision Letter · Decision Letter 0]

29 Dec 2020

PONE-D-20-26801

Early detection of complications in pancreas transplants by microdialysis catheters, an observational feasibility study.

PLOS ONE

Dear Dr. Kjøsen,

Thank you for submitting your manuscript to PLOS ONE. After careful consideration, we feel that it has merit but does not fully meet PLOS ONE’s publication criteria as it currently stands. Therefore, we invite you to submit a revised version of the manuscript that addresses the points raised during the review process.

Apart from the points made by the reviewers I would appreciate if your would address the following points: firstly, the trial registration number provided in the manuscript concerns a different study protocol, secondly it would be helpful if to the readers if the reason for early termination in mentioned in the study, finally I am missing a possible sampling delay which obviously involved the dead-volume of the system and the volume required for the analyses.

We look forward to receiving your revised manuscript.

Kind regards,

Jan H.N. Lindeman

Academic Editor

PLOS ONE

Journal Requirements:

2.)Thank you for stating the following in the Funding Section of your manuscript:

"No"

 "This work was supported by an unrestricted research grant from South-Eastern Norway Regional Health Authority (2016028)"

3.) Thank you for stating the following in your Competing Interests section: 

'No'

4.) We note that you have indicated that data from this study are available upon request. PLOS only allows data to be available upon request if there are legal or ethical restrictions on sharing data publicly. For information on unacceptable data access restrictions, please see http://journals.plos.org/plosone/s/data-availability#loc-unacceptable-data-access-restrictions.

5.) Please amend your list of authors on the manuscript to ensure that each author is linked to an affiliation. Authors’ affiliations should reflect the institution where the work was done (if authors moved subsequently, you can also list the new affiliation stating “current affiliation:….” as necessary).

6.) Your ethics statement should only appear in the Methods section of your manuscript. If your ethics statement is written in any section besides the Methods, please delete it from any other section.

7.) Please include captions for your Supporting Information files at the end of your manuscript, and update any in-text citations to match accordingly. Please see our Supporting Information guidelines for more information: http://journals.plos.org/plosone/s/supporting-information.

8.) Please ensure that you refer to Figure 4 in your text as, if accepted, production will need this reference to link the reader to the figure.

Reviewers' comments:

Reviewer's Responses to Questions

**Comments to the Author**

1. Is the manuscript technically sound, and do the data support the conclusions?

Reviewer #1: Yes

Reviewer #2: Yes

2. Has the statistical analysis been performed appropriately and rigorously? 

Reviewer #1: Yes

Reviewer #2: Yes

3. Have the authors made all data underlying the findings in their manuscript fully available?

Reviewer #1: No

Reviewer #2: Yes

4. Is the manuscript presented in an intelligible fashion and written in standard English?

Reviewer #1: Yes

Reviewer #2: Yes

5. Review Comments to the Author

Reviewer #1: A single group feasibility study was conducted to detect early complications in pancreas transplants (n=34) by monitoring glucose, lactate, pyruvate, and glycerol levels every 1-2 hours post transplant. The median lactate in these patients was significantly higher in both catheters compared to non-events (n=15). The statistical methods were clearly written.

Minor revisions:

1- Line 156: Indicate the statistical testing method which achieves 80% power.

2- Express p-values more precisely, rather than p>0.05.

Reviewer #2: - In methods, line 126: It is not clear when the second measurement is performed. Please specify.

- In methods, line 132: What was the imaging protocol before the ninth patient in the study?

- In results, line 196: It would be interesting to know when the loss of functional sampling occurred.

- In results: Was there any complication related to microdyalisis catheters placement?

- In discussion: It could be interesting to discuss whether one determination is enough to indicate CT to gain time in thrombus diagnosis.

- In discussion: It could be interesting to know if there were differences in donor characteristics between groups.

6. PLOS authors have the option to publish the peer review history of their article (what does this mean?). If published, this will include your full peer review and any attached files.

Reviewer #1: No

Reviewer #2: No

---

## [Author Response · Author response to Decision Letter 0]

3 Feb 2021

Dear Dr. Lindeman

Thank you so much for your invitation to resubmit our manuscript. We wish to thank both you, and the reviewers for your time, and for the generous and invaluable feedback. We have edited the manuscript to address the questions and concerns raised.

Below, we will address each question separately.

Apart from the points made by the reviewers I would appreciate if your would address the following points: firstly, the trial registration number provided in the manuscript concerns a different study protocol, secondly it would be helpful if to the readers if the reason for early termination in mentioned in the study, finally I am missing a possible sampling delay which obviously involved the dead-volume of the system and the volume required for the analyses.

Unfortunately, this detail was not presented clearly in the original manuscript. The trial registration number is indeed the correct one. The submitted manuscript is however, a substudy of a larger one pertaining to pancreas transplants (The Norwegian Pancreas Transplantation (PTx) Study. The details regarding the microdialysis part can be found on clinicaltrials.gov (NCT01957696), under “other outcome measures”, #6 and #7. 

Changes made accordingly in manuscript pg. 4, lines 94-95.

Thank you for allowing us to clarify the reason for the early termination of the study. As stated in the manuscript, the study was terminated prematurely due to an unforeseen increase in the incidence of thrombi. Our pre-study incidence was around 10%, and by the time of termination this number was 26%. There are several possible explanations for this, of which study bias is the most obvious. We actively tried to find thrombi by close monitoring with microdialysis and routine CT scans. Some of the thrombi were located to the distal splenic vein, these are thrombi that usually does not require any intervention. Most likely, some thrombi would also have gone unnoticed without a CT scan. The catheters are not likely to be the cause. Our research group have used microdialysis in a variety of settings and have implanted hundreds of these catheters without complications. In this paper the catheters were also external to the pancreatic graft, which makes it challenging to understand physiologically, how this could lead to thrombi in the graft itself. 

Changes made accordingly in manuscript pg. 20, lines 396-401.

There is unarguably a sampling delay in the microdialysis setup. According to the manufacturer the membrane-to-collection delay is approximately 8 minutes at a rate of 1 µL/min. The analysis itself takes about 10 minutes to complete. Therefore, the total delay is about 18 minutes. In the presence of a venous thrombus, we feel this delay is not the decisive factor in terms of outcome.

Changes made accordingly in manuscript pgs. 5-6, lines 125-126

Reviewer #1: Minor revisions:

1- Line 156: Indicate the statistical testing method which achieves 80% power.

Thank you for alerting us to this detail which escaped the manuscript. 

For sample size calculation, we used Power Analysis (A& X Analytics LLC, Boston, MA) adjusted for later test with an unpaired two sample two tail t-test. We have added this sentence to the manuscript and the information about the parameters of the power calculation, pg. 7, lines 160-165.

2- Express p-values more precisely, rather than p>0.05.

Thank you, there was one p-value that “fell through” the read-through. It has now been corrected to the exact p-value of 0.26. 

Changes made accordingly in manuscript pg. 16, line 284.

Reviewer #2: 

- In methods, line 126: It is not clear when the second measurement is performed. Please specify.

The repeat measurement was done 30 minutes later. The choice of performing a second measurement was to lessen the likelihood of the elevation representing a spurious measurement.

Changes made accordingly in manuscript pg. 6, lines 131-132.

- In methods, line 132: What was the imaging protocol before the ninth patient in the study?

Before the ninth patient, the only mandatory radiological imaging, was an ultrasound performed within four hours after reperfusion. CT scan was only performed if the ultrasonographic images was of inadequate quality for determining arterial and venous flow. 

Changes made accordingly in manuscript pg. 6, lines 136-137.

- In results, line 196: It would be interesting to know when the loss of functional sampling occurred.

The 17 catheters that ceased functioning, had a median catheter life of 5,5 days for the anterior catheter (IQR 4-7, range 1-13), and 5 days for the posterior catheter (IQR 3.8-6.5, range 1-13). In our opinion, this range gives a “practical” approach to describe the catheter lifetime. The loss of functional sampling is most likely due to clogging of the microdialysis membrane by the protein rich exudate found surrounding pancreas grafts.

Changes made accordingly in manuscript pg. 11, lines 205-207.

- In results: Was there any complication related to microdyalisis catheters placement?

Thank you for allowing us to address this question. We have after more than a decade and several hundreds of microdialysis catheters placed, yet to encounter significant complications both upon insertion and removal. In these 34 patients there were no complications on insertion whatsoever. 

Changes made accordingly in manuscript pg. 11, lines 207-208.

- In discussion: It could be interesting to discuss whether one determination is enough to indicate CT to gain time in thrombus diagnosis.

The microdialysis method, like any method of measurement is inherently associated with margins of error, false positives, and negatives. Despite calibration and quality controls, outliers of measurements still occur. A false positive test would, in the instance of a single determination cut off, lead to a great number of repeat CT scans. Aside from the cost and other logistical factors, this would mean a significant risk exposure both in terms of radiation, and that of nephrotoxic contrasting agents necessary for adequate imaging of the pancreatic vessels. In this study, 18/32 transplantations were by means of simultaneous pancreas kidney transplantation. In most other countries this ratio is towards 3 in 4, or higher. Using nephrotoxic contrasting agents in this group, as with any examination, should be restricted to cases where the benefit of the examination outweighs the risk. The one pretest modifier we can employ here to aid in this selection, is to minimize the false positives. In this instance, repeating the measurement. Yes, the time to diagnosis might be somewhat shortened by relying on one determination, but not without imposing a significant increased risk to the patient. We have added a paragraph in the discussion, pg. 20, lines 385-395.

- In discussion: It could be interesting to know if there were differences in donor characteristics between groups.

There were indeed some differences in the characteristics of the subpopulations. With three pathological cohorts and one control group, we performed a subgroup analysis. This revealed a statistically significant lower total arterial pancreatic blood flow, and a shorter cold ischaemia time in the thrombus group versus the reference cohort. Identical differences were found between the haematoma group versus the reference cohort. In the discussion of the revised manuscript (pgs. 18-19, lines 350-367), we have posed hypotheses for the possible explanations for these findings. However, the event subgroups were small in size, and care should be exercised in drawing conclusions based solely on these data.

Again, we would like to thank the editor and referees for their review and evaluation. We have tried to address all the concerns raised in an adequate manner, and believe that by answering these, the manuscript has improved considerably. We would however be happy to make further corrections if this is necessary. We look forward to hearing back from you soon.

Sincerely,

Gisle Kjøsen, MD

Consultant Anaesthesiologist

Department of Emergencies and Critical Care Medicine

Oslo University Hospital, Rikshospitalet

Norway

---

## [Editor Report · Decision Letter 1]

10 Feb 2021

Early detection of complications in pancreas transplants by microdialysis catheters, an observational feasibility study.

PONE-D-20-26801R1

Dear Dr. Kjøsen,

We’re pleased to inform you that your manuscript has been judged scientifically suitable for publication and will be formally accepted for publication once it meets all outstanding technical requirements.

Kind regards,

Jan H.N. Lindeman

Academic Editor

PLOS ONE
---

## [Editor Report · Acceptance letter]

15 Feb 2021

PONE-D-20-26801R1 

Early detection of complications in pancreas transplants by microdialysis catheters, an observational feasibility study. 

Dear Dr. Kjøsen:

I'm pleased to inform you that your manuscript has been deemed suitable for publication in PLOS ONE. Congratulations! Your manuscript is now with our production department. 

Kind regards, 

on behalf of

Dr. Jan H.N. Lindeman 

Academic Editor

PLOS ONE